# A unique van Hove singularity in kagome superconductor $CsV_{3-x}Ta_xSb_5$ with enhanced superconductivity

Yang Luo[1,9], Yulei Han [2,9], Jinjin Liu[3,4,9], Hui Chen [5,9], Zihao Huang[5], Linwei Huai[1], Hongyu Li[1], Bingqian Wang[1], Jianchang Shen[1], Shuhan Ding[1], Zeyu Li[1], Shuting Peng [1], Zhiyuan Wei[1], Yu Miao[1], Xiupeng Sun[1], Zhipeng Ou[1], Ziji Xiang [1], Makoto Hashimoto [6], Donghui Lu [6], Yugui Yao [3,4], Haitao Yang [5], Xianhui Chen [1], Hong-Jun Gao [5] ✉, Zhenhua Qiao [1,7] ✉, Zhiwei Wang [3,4,8] ✉ & Junfeng He [1] ✉

Van Hove singularity (VHS) has been considered as a driving source for unconventional superconductivity. A VHS in two-dimensional (2D) materials consists of a saddle point connecting electron-like and hole-like bands. In a rare case, when a VHS appears at Fermi level, both electron-like and hole-like conduction can coexist, giving rise to an enhanced density of states as well as an attractive component of Coulomb interaction for unconventional electronic pairing. However, this van Hove scenario is often destroyed by an incorrect chemical potential or competing instabilities. Here, by using angle-resolved photoemission measurements, we report the observation of a VHS perfectly aligned with the Fermi level in a kagome superconductor $CsV_{3-x}Ta_xSb_5$ ($x \sim 0.4$), in which a record-high superconducting transition temperature is achieved among all the current variants of $AV_3Sb_5$ (A = Cs, Rb, K) at ambient pressure. Doping dependent measurements reveal the important role of van Hove scenario in boosting superconductivity, and spectroscopic-imaging scanning tunneling microscopy measurements indicate a distinct superconducting state in this system.

In correlated materials, the VHS sets a paradigm to understand the divergence of electron density of states. At the VHS, Coulomb interactions between electrons may significantly exceed their kinetic energy, which would in turn drive multiple instabilities in the system, including charge density wave (CDW), spin density wave (SDW), $s/d$-wave superconductivity[1–4]. In the context of superconductivity, an ideal van Hove scenario[5] works primarily via two routes. One is to substantially increase the density of states for Cooper pairs mediated by either

[1]Department of Physics and CAS Key Laboratory of Strongly-coupled Quantum Matter Physics, University of Science and Technology of China, Hefei, Anhui 230026, China. [2]Department of Physics, Fuzhou University, Fuzhou, Fujian 350108, China. [3]Centre for Quantum Physics, Key Laboratory of Advanced Optoelectronic Quantum Architecture and Measurement (MOE), School of Physics, Beijing Institute of Technology, Beijing 100081, China. [4]Beijing Key Lab of Nanophotonics and Ultrafine Optoelectronic Systems, Beijing Institute of Technology, Beijing 100081, China. [5]Beijing National Center for Condensed Matter Physics and Institute of Physics, Chinese Academy of Sciences, Beijing 100190, China. [6]Stanford Synchrotron Radiation Lightsource, SLAC National Accelerator Laboratory, Menlo Park, CA 94025, USA. [7]International Center for Quantum Design of Functional Materials, University of Science and Technology of China, Hefei, Anhui 230026, China. [8]Material Science Center, Yangtze Delta Region Academy of Beijing Institute of Technology, Jiaxing 314011, China. [9]These authors contributed equally: Yang Luo, Yulei Han, Jinjin Liu, Hui Chen. ✉e-mail: hjgao@iphy.ac.cn; qiao@ustc.edu.cn; zhiweiwang@bit.edu.cn; jfhe@ustc.edu.cn

conventional or unconventional electron-boson coupling[5]; the other is to form an unconventional electronic pairing, driven by an attractive component of the Coulomb interaction from the coexisted electrons and holes at the VHS[5]. However, the above scenario has a stringent requirement on the precise energy alignment between the VHS and the Fermi level ($E_F$) due to the logarithmical divergence of the associated density of states[5], making its realization in real materials challenging.

Recently, the $AV_3Sb_5$ kagome metals have attracted a lot of attention[6–44] as the first material family to realize superconductivity in layered kagome systems[7–9]. In particular, the existence of multiple VHSs in the electronic structure of $AV_3Sb_5$[10–13] provides a fertile territory to explore the possible manifestation of van Hove scenario[5] in kagome materials. This expectation is also in accord with the phase diagram of $AV_3Sb_5$, which is primarily dominated by CDW and superconductivity[7–9,14–23]—two leading instabilities of the 2D VHS. However, despite the exciting theoretical proposals[1–4,24–27], the direct experimental evidence to illustrate the role of VHS in this system is still lacking.

In this work, we investigate this issue by comparing the pristine $CsV_3Sb_5$ with our newly discovered $CsV_{3-x}Ta_xSb_5$ samples. With Ta substitution (Fig. 1a, b), the CDW order is suppressed and the superconducting transition temperature ($T_c$) is doubled from ~2.5 Kelvin (K) in $CsV_3Sb_5$ to ~5.5 K in $CsV_{3-x}Ta_xSb_5$ with $x \sim 0.4$, which is the highest among all the current variants of $AV_3Sb_5$ at ambient pressure[7–9,21,22,28–30].

Angle-resolved photoemission spectroscopy (ARPES) measurements on the new compound $CsV_{3-x}Ta_xSb_5$ ($x \sim 0.4$) reveal a perfect realization of the VHS at Fermi level, formed by V $d$-orbitals of the kagome lattice. Surprisingly, the special VHS is almost quantitatively reproduced by first-principles calculations considering the Ta substitution. This is distinct from that of pristine $CsV_3Sb_5$, where the VHSs are pushed away from the Fermi level by the CDW order. We further demonstrate that the suppression of competing orders (e.g., the CDW order) is insufficient to account for the record-high $T_c$, and the superconducting $T_c$ (gap) is related to the energy position of the VHS in samples without competing orders. In the meantime, negligible changes are observed on other low-energy states and their associated electron-boson coupling as a function of Ta substitution. As such, our results establish a direct link between the substantially enhanced superconductivity and the VHS at the Fermi level. Our spectroscopic-imaging scanning tunneling microscopy (STM) measurements further reveal that the superconducting state of $CsV_{3-x}Ta_xSb_5$ ($x \sim 0.4$) is different from that of the pristine $CsV_3Sb_5$. These results demonstrate the feasibility of van Hove scenario[5] in kagome superconductors.

## Results
### Fermi surface and electronic structure

We start from investigating the Fermi surfaces of $CsV_3Sb_5$ and $CsV_{2.6}Ta_{0.4}Sb_5$ (Fig. 1c, d). The overall Fermi surface topology remains

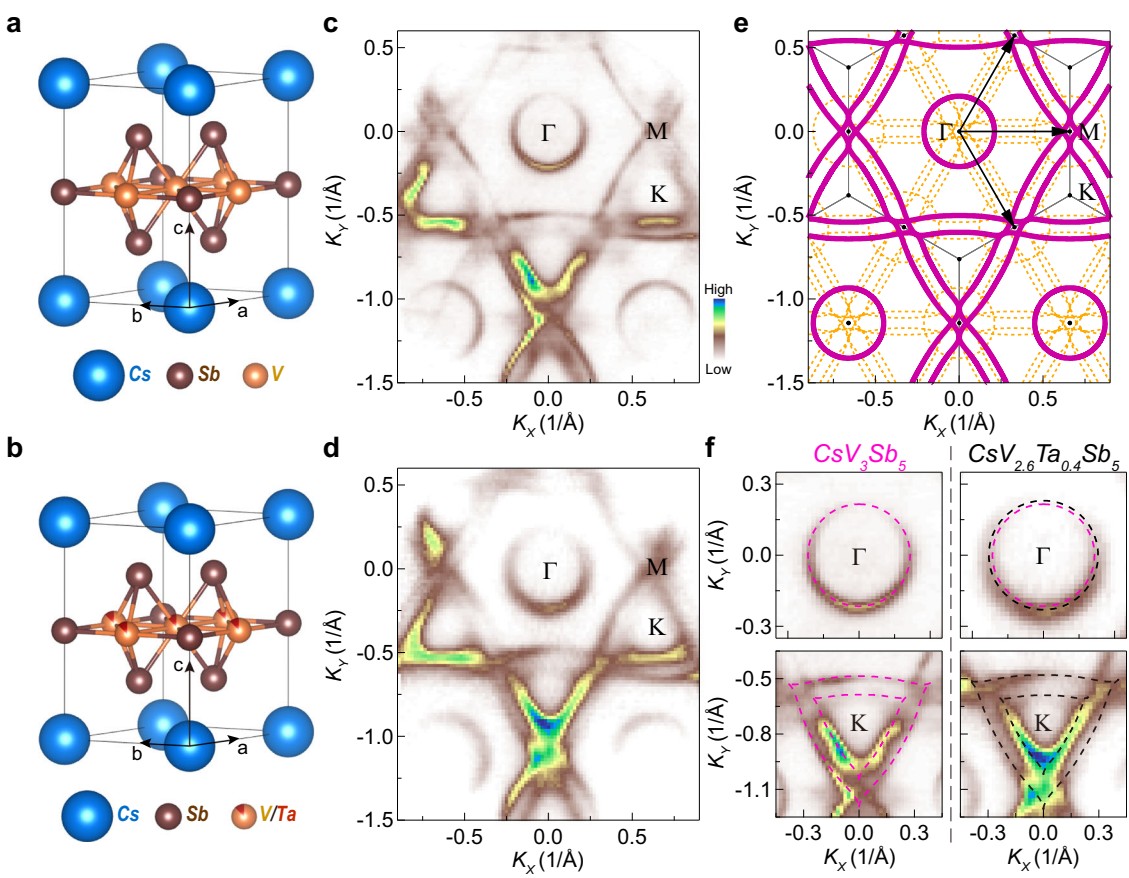

**Fig. 1 | Fermi surface of $CsV_3Sb_5$ and Ta substituted $CsV_3Sb_5$ measured at low temperature (25 K). a, b** Crystal structure of $CsV_3Sb_5$ (**a**) and Ta substituted $CsV_3Sb_5$ (**b**). **c, d** Fermi surface of $CsV_3Sb_5$ (**c**) and $CsV_{2.6}Ta_{0.4}Sb_5$ (**d**) measured with 56 eV photons, which probe the electronic structure in the Γ-K-M plane[33] (Supplementary Fig. S2). Characteristic features are identified in both compounds, including an electron-like Fermi pocket around the Γ point, double-triangular Fermi surface sheets centered at the K point and their shared corners at the M point. **e** Schematic of the original Fermi surface (solid magenta line), and folded Fermi surface (orange dash line) by the CDW order. The in-plane wavevectors of the CDW are shown by black arrows. **f** Fermi surface sheets around Γ and K points of $CsV_3Sb_5$ and $CsV_{2.6}Ta_{0.4}Sb_5$, respectively. Same as those in (**c**) and (**d**), but shown in an expanded scale. The dashed lines are a guide for the eye. The same magenta dashed circle is appended to both Fermi surface sheets around Γ point for a quantitative comparison. The electron-like Fermi pocket becomes slightly larger in the $CsV_{2.6}Ta_{0.4}Sb_5$ sample.

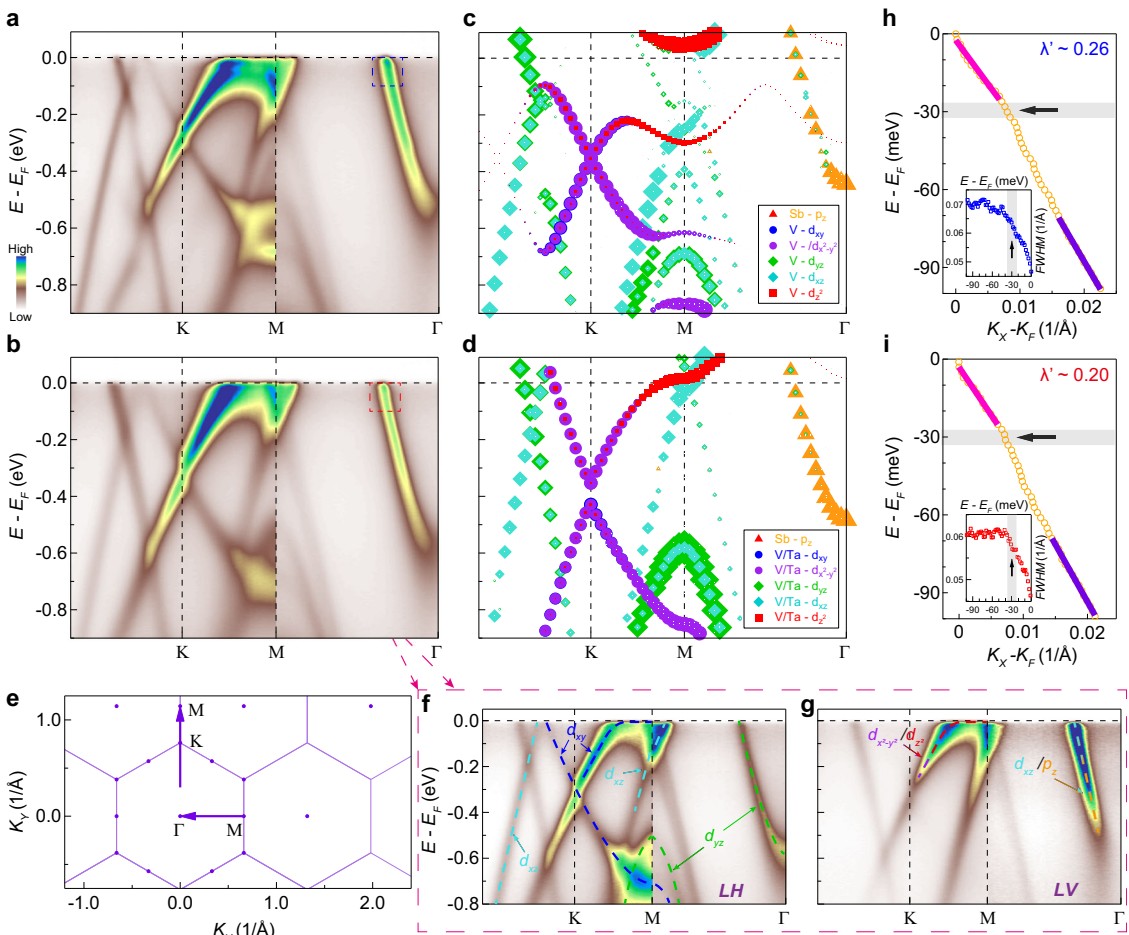

**Fig. 2 | Band structure of CsV₃Sb₅ and Ta substituted CsV₃Sb₅ measured at low temperature (25 K). a, b** Photoelectron intensity plots along Γ-K-M-Γ of CsV₃Sb₅ (**a**) and CsV$_{2.6}$Ta$_{0.4}$Sb₅ (**b**) measured with 56 eV circularly polarized photons. **c, d** Calculated orbital-resolved band structure along Γ-K-M-Γ of CsV₃Sb₅ (**c**) and Ta substituted CsV₃Sb₅ (**d**). Two Ta atoms are considered in a 2 × 2 supercell to simulate the Ta substituted sample. Different orbitals are marked by different colors. The size of the markers represents the spectral weight of the orbitals. The electron-like band near Γ mainly consists of the Sb $p_z$ orbital, the Dirac cone around K is primarily contributed by V/Ta $d_{xy}$ and $d_{x^2-y^2}$ orbitals, and the low energy VHS near M is dominated by the V/Ta $d_{z^2}$ orbital with contributions from the V/Ta $d_{xy}$

orbital. **e** Schematic of the BZ in the Γ-K-M plane. Arrows indicate the locations of the momentum cuts. **f, g** Photoelectron intensity plots along Γ-K-M-Γ of CsV$_{2.6}$Ta$_{0.4}$Sb₅ measured with 56 eV linear horizontally (LH) polarized (**f**) and linear vertically (LV) polarized (**g**) light. Dashed lines in (**f**), (**g**) are the eye-guide for orbitals probed by each polarization, respectively. **h** MDC-derived dispersion of the electron-like band around Γ in the boxed area in (**a**). λ´ marks the effective coupling strength. The ratio between the high-energy velocity above the kink energy (purple) and the dressed velocity below the kink energy (pink) is defined as λ´+1. Full-width-at-half-maximum (FWHM) of the MDC peaks is shown in the inset. **i** Same as (**h**), but extracted from the CsV$_{2.6}$Ta$_{0.4}$Sb₅ data in (**b**).

similar in both materials. Some additional Fermi surface sheets are seen in the pristine CsV₃Sb₅ with relatively weak spectral intensity (Fig. 1c and Supplementary Fig. S1), which represent a replica of the main Fermi surface (Fig. 1e). This Fermi surface folding effect is believed to be associated with the CDW order[31,32], which gives rise to a scattering wavevector connecting M-M (or equivalently Γ-M) in the Brillouin zone (BZ) (Fig. 1e). On the contrary, the folding induced replica Fermi surface sheets are absent in CsV$_{2.6}$Ta$_{0.4}$Sb₅ (Fig. 1d), which is consistent with the complete suppression of the CDW order in this Ta substituted sample. In this regard, we compare the bands of CsV$_{2.6}$Ta$_{0.4}$Sb₅ with the main bands of the pristine CsV₃Sb₅, hereafter.

Similar to the Fermi surface, the overall band structure of CsV$_{2.6}$Ta$_{0.4}$Sb₅ also shares the characteristic features with the main bands of the pristine CsV₃Sb₅, evidenced by an electron-like band centered at the Γ point, a Dirac crossing at the K point and multiple VHSs near the M point (Fig. 2a, b). First-principles calculations have been carried out on Ta substituted CsV₃Sb₅ (see Supplementary Figs. S3 and S4 for details). The corresponding orbital characters have been analyzed in the calculations and confirmed by polarization-dependent photoemission measurements (Fig. 2d, f, g,

Supplementary Figs. S5–S7), which are similar to those identified in the pristine CsV₃Sb₅[13,33]. The persistence of the overall band structure and the associated orbital characters demonstrates the robustness of the kagome lattice upon Ta substitution, which also enables a quantitative examination of the changes in low energy states at different momenta.

For this purpose, we first examine the simple electron-like band centered at the Γ point. In the pristine CsV₃Sb₅, a dispersion kink at ~30 meV is observed on this band due to the improved data quality (Fig. 2h), indicating the existence of electron-boson coupling in the Sb $p$-states. This band forms an electron-like pocket on the Fermi surface, which remains gapless irrespective of the CDW order[31,34,35]. In CsV$_{2.6}$Ta$_{0.4}$Sb₅, the electron-like band moves towards a slightly deeper binding energy from the E$_F$ (Fig. 2a, b), which is also captured by first-principles calculations (Fig. 2c, d). The dispersion kink remains at ~30 meV, with a similar or smaller coupling strength (e.g., compare Fig. 2h, i, also see Supplementary Fig. S8). The corresponding electron-like Fermi pocket becomes slightly larger (Fig. 1f), and remains gapless (Figs. 3a and 4a). Then, we examine the electronic structure near the K point. The Dirac crossing moves towards a

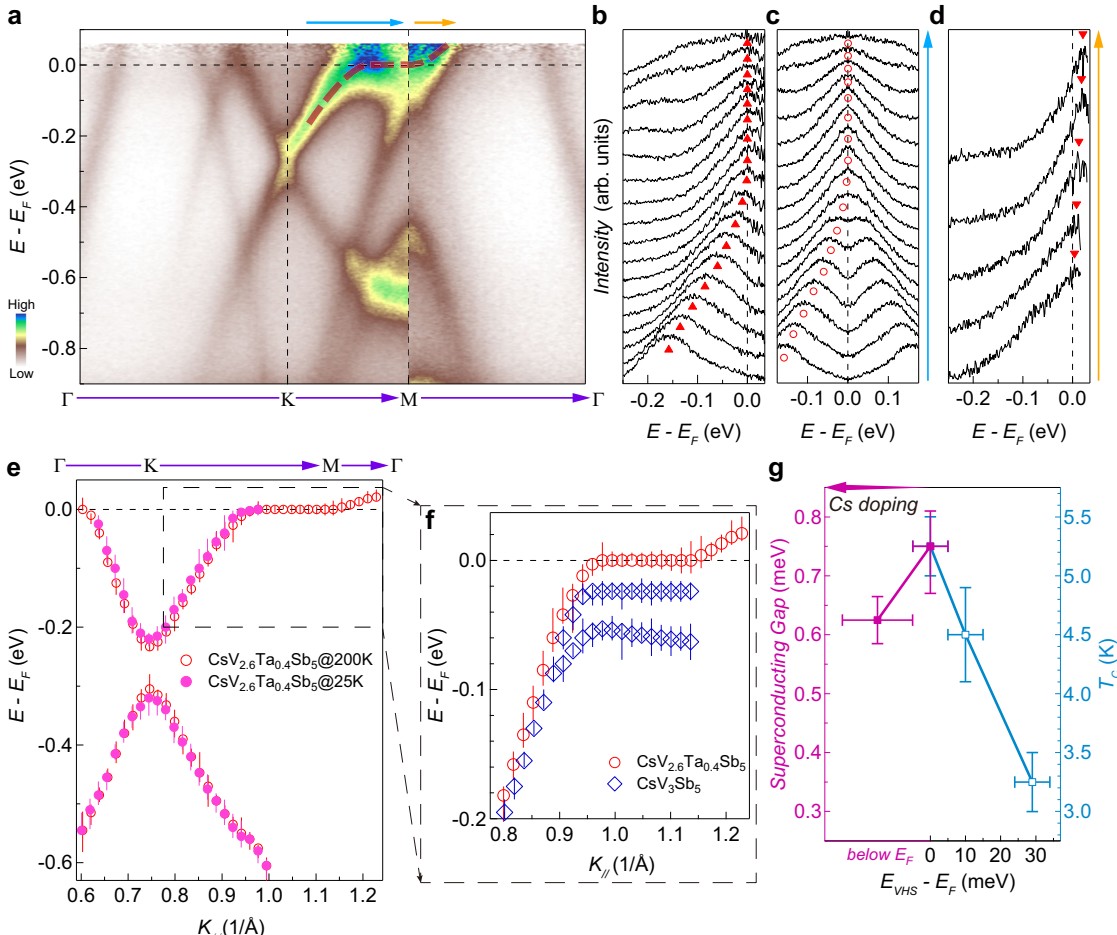

**Fig. 3 | The existence of VHS perfectly aligned with the Fermi level in CsV$_{2.6}$Ta$_{0.4}$Sb$_5$. a** Photoelectron intensity plot along Γ-K-M-Γ of CsV$_{2.6}$Ta$_{0.4}$Sb$_5$ measured with 21.2 eV photons at 200 K. This photon energy probes the electronic structure in the Γ-K-M plane of the 3D BZ (Supplementary Fig. S2). The red dashed lines are a guide for the eye. **b** EDCs near M point extracted from the photoemission raw spectrum in the momentum region marked by the blue arrow in (**a**). **c** Same as the raw EDCs in (**b**), but symmetrized to show the absence of an energy gap. **d** EDCs near M point extracted from the photoemission raw spectrum in the momentum region marked by the yellow arrow in (**a**). The red triangles and circles in (**b**), (**d**) indicate the EDC peaks. **e** Band dispersion near K and M points (along the Γ-K-M-Γ direction) extracted from the EDC peaks measured at 200 K (red empty circles) and 25 K (magenta solid circles), respectively. **f** Extracted band dispersion of CsV$_{2.6}$Ta$_{0.4}$Sb$_5$ and CsV$_3$Sb$_5$ in the momentum region near M, indicated by the black dotted box in (**e**). The error bars in (**e**) and (**f**) represent the uncertainties in the determination of EDC peak positions. **g** Superconducting $T_c$ (right axis) and superconducting gap (left axis) as a function of the energy position of the VHS in doped CsV$_3$Sb$_5$ samples (see Supplementary Figs. S12–S14). The error bars in (**g**) represent the uncertainties in the determination of the VHS (bottom axis), superconducting $T_c$ (right axis), and superconducting gap (left axis).

slightly deeper binding energy (Fig. 2a, b), but the low energy states between Γ and K (Fig. 2a, b) as well as the area of the double-triangle Fermi surface sheets around the K point exhibit little change upon the Ta substitution (Fig. 1f).

The most significant changes take place near the M point. In the pristine CsV$_3$Sb$_5$, electronic structures in this momentum region are significantly reconstructed by the CDW order[10–13,36], giving rise to an additional VHS at -70 meV below E$_F$ (Figs. 2a and 3f)[12]. The low energy states near M are also gapped away from E$_F$ by the CDW order, with a gap size of ~20 meV (Fig. 3f and Supplementary Fig. S9)[31,34,35]. Distinct from that in the pristine CsV$_3$Sb$_5$, the CDW induced VHS and energy gap are absent in CsV$_{2.6}$Ta$_{0.4}$Sb$_5$ (Fig. 2b and Supplementary Fig. S9), and the electronic structure remains identical at 200 K and 25 K (Fig. 3e). This is also consistent with the absence of CDW order in this compound. Surprisingly, a careful examination of the VHS near the M point of CsV$_{2.6}$Ta$_{0.4}$Sb$_5$ reveals a flat dispersion exactly at the Fermi level, which connects the top of a hole-like band to the bottom of an electron-like band (Fig. 3a, Supplementary Figs. S10 and S11). The energy position of the VHS can be quantitatively determined by the energy distribution curves (EDCs) of the raw data. The Fermi-Dirac

Function is removed to observe the band structure slightly above the Fermi level, and to avoid the shift of the EDC peaks by the Fermi-Dirac distribution. As shown in Fig. 3b, the hole-like band disperses towards the Fermi level and merges into a flat dispersion. The quasiparticle peaks in the flat dispersion region are unambiguously identified at the Fermi level without any energy gap (Fig. 3b, c). At the end of the flat dispersion, a small portion of the connected electron-like band can be seen above the Fermi level due to the thermal broadening (Fig. 3d). This is a standard example of a perfect VHS at the Fermi level, where the hole-like conduction and electron-like conduction coexist. It is interesting to note that such a VHS is almost quantitatively reproduced by first-principles calculations considering the Ta substitution (Fig. 2d).

### Superconducting state

After revealing the electronic structure by ARPES, we investigate the superconducting state by STM. The superconducting gap of CsV$_{2.6}$Ta$_{0.4}$Sb$_5$ shows a U-like shape, where the gap gradually forms between −0.8 meV and −0.4 meV, and exhibits nearly zero conductance in the energy region near E$_F$ (Fig. 4e, g). This is distinct from

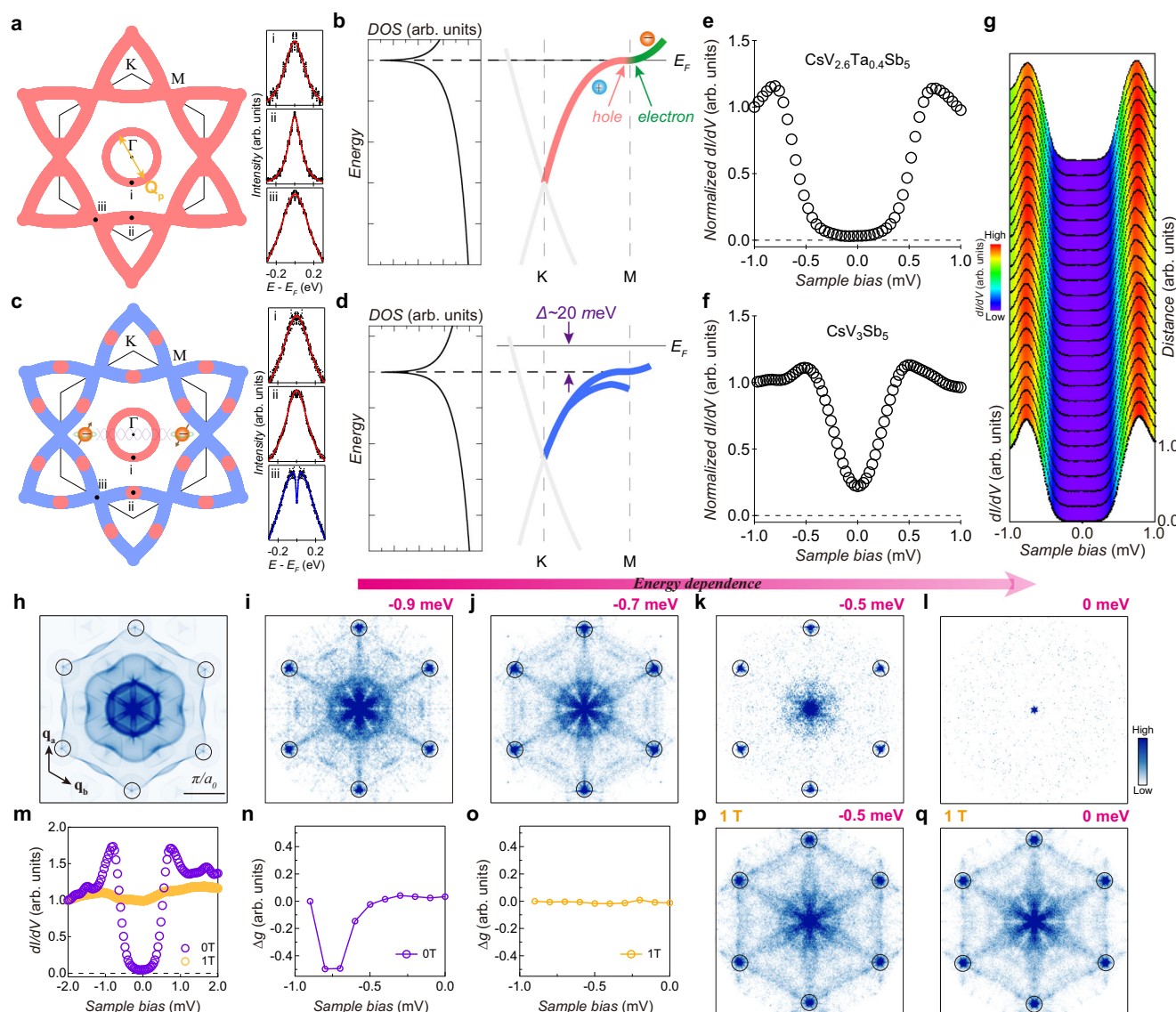

**Fig. 4 | Distinct superconducting state in CsV$_{2.6}$Ta$_{0.4}$Sb$_5$. a** Schematic of the Fermi surface of CsV$_{2.6}$Ta$_{0.4}$Sb$_5$. The gapless (gapped) momentum regions are shown in red (blue). Symmetrized EDCs at selected momentum points (marked by i, ii, and iii) are presented. The overlaid curves represent the fitting results by the Norman function. The orange arrow indicates the scattering wavevector $\mathbf{Q_p}$ connecting the circular Fermi pocket around $\Gamma$. **b** Schematic of the band structure along the K-M direction and the electron density of states associated with the VHS. **c, d** Same as (**a**), (**b**), but for the pristine CsV$_3$Sb$_5$. **e, f** Spatially averaged $dI/dV$ spectrum of CsV$_{2.6}$Ta$_{0.4}$Sb$_5$ (**e**) and CsV$_3$Sb$_5$ (**f**) measured at 0.4 K. **g** Waterfall-like plot of the $dI/dV$ spectra, showing a uniform superconducting gap along a line-cut on CsV$_{2.6}$Ta$_{0.4}$Sb$_5$. **h** Simulated QPI pattern calculated by the autocorrelation of the approximate schematic of the Fermi surface. **i–l** Six-fold symmetrized Fourier transform of $dI/dV$ maps measured on the Sb surface of CsV$_{2.6}$Ta$_{0.4}$Sb$_5$ at 0.4 K, with an energy of −0.9 meV, −0.7 meV, −0.5 meV, and 0 meV, respectively. **m** $dI/dV$ spectra measured at 0 Tesla (T) and 1 T. **n** $\Delta g$ plotted as a function of energy. **o** Same as (**n**), but measured with a magnetic field of 1 T. **p, q** Same as (**k**), (**l**), but measured with a magnetic field of 1 T. The unsymmetrized raw data of the Fourier transform of $dI/dV$ maps and STM setup conditions are shown in Supplementary Fig. S16.

that in the pristine CsV$_3$Sb$_5$ sample, where a V-shape superconducting gap is identified[15] (Fig. 4f). To better understand the superconducting state of CsV$_{2.6}$Ta$_{0.4}$Sb$_5$, quasiparticle interference (QPI) measurements have been carried out. When the energy is beyond the superconducting gap (e.g., −0.9 meV, Fig. 4i), the Fourier transform of the measured QPI image shows a clear resemblance to the simulation by autocorrelation of the proximate schematic of the Fermi surface (Fig. 4h), where scattering patterns from both V $d$-orbitals and Sb $p$-orbitals can be clearly identified. For example, the scattering wavevector $\mathbf{Q_p}$ connecting the circular Fermi pocket around $\Gamma$ (Sb $p$-orbitals, Fig. 4a) gives rise to a circle at the center of the scattering pattern (Fig. 4h), and the scattering wavevectors connecting Fermi surface sheets with V $d$-orbitals give rise to multiple flower-shape scattering patterns (Fig. 4h). It seems that the scattering patterns from the V $d$-

orbitals vanish more rapidly than that from the Sb $p$-orbitals between −0.8 meV and −0.4 meV (Fig. 4i–k). In order to quantify the difference, QPI intensities for the V orbitals and Sb orbitals are integrated, respectively. The difference of the normalized QPI intensities $\Delta g$ is shown as a function of energy (Fig. 4n, see Supplementary Fig. S15). This difference disappears when the superconductivity is suppressed by an external magnetic field (Fig. 4o). These results indicate that the V $d$-orbitals are more strongly suppressed than Sb $p$-orbitals between −0.8 meV and −0.4 meV. On the other hand, the V $d$-orbitals in the pristine CsV$_3$Sb$_5$ are primarily gapped by the CDW order, whereas the Sb $p$-orbitals remain gapless in the CDW state (Fig. 4c and Supplementary Fig. S9)[31,34,35]. Therefore, when the CsV$_3$Sb$_5$ enters the superconducting state from the CDW state, one would naturally expect that the gapless electron-like pocket with Sb $p$-orbitals can provide electron

density of states for the pairing process (Fig. 4c). QPI measurements indicate that the Sb $p$-orbitals are involved in the superconductivity, although they may be influenced by multiple other orders in the pristine compound (Supplementary Figs. S17 and S18). We cannot rule out that the remnant V $d$-electrons inside the CDW gap may also contribute to Cooper pairs, but the VHS at ~20 meV below $E_F$ near the M point has little effect on the superconductivity (Supplementary Fig. S19).

## Discussion

Next, we discuss the implications of our observations. A direct question is about the origin of the enhanced superconductivity in the Ta substituted compound. In principle, the substitution of V by Ta might induce a chemical strain in the sample. Nevertheless, we find that the substantially enhance $T_c$ is special for the Ta substitution, which is not a universal property in the $CsV_3Sb_5$ system with the similar chemical strain (Supplementary Fig. S20). Therefore, we examine possible contributions from the unique electronic structure of the $CsV_{2.6}Ta_{0.4}Sb_5$ sample. First, the electron-like band with Sb $p$-orbitals around Γ has little contribution to the substantially increased $T_c$. The electron-boson coupling on this band remains similar or becomes slight weaker with the Ta substitution (Fig. 2h, i). Second, the low energy states between the Γ and K points may potentially involve the pairing process, but they are almost identical in the $CsV_3Sb_5$ and $CsV_{2.6}Ta_{0.4}Sb_5$ samples (Figs. 2a, b and 4a, c). Distinct from the above two observations, the superconducting $T_c$ (gap) shows a clear relationship with the energy position of the VHS in various doped samples without competing orders, where the maximum $T_c$ (gap) is associated with the VHS at $E_F$ [Fig. 3g, also see Supplementary Figs. S12–S14 for the comparison between a Ti substituted sample $CsV_{3-x}Ti_xSb_5$ ($x$ ~ 0.2), two Ta substituted samples $CsV_{3-x}Ta_xSb_5$ ($x$ ~ 0.3 and $x$ ~ 0.4) and surface doped samples]. These results have experimentally demonstrated that the suppression of competing orders in Ta substituted samples is insufficient to account for the record-high $T_c$, and a direct connection is established between the significantly enhanced superconductivity and the appearance of VHS at the Fermi level in the Ta substituted sample. Theoretically, a VHS perfectly aligned with the Fermi level can remarkably change the superconductivity at least via two routes. In a more traditional picture, where the Cooper pairs are formed by a bosonic pairing mechanism, the substantially enhanced density of states at the Fermi level by the VHS (Fig. 4b) would significantly enhance the superconductivity and increase the $T_c$ (Supplementary Fig. S19). This cooperative mechanism between the bosonic pairing and the van Hove scenario[5] works for both conventional electron–phonon coupling and other unconventional bosonic pairing. The second route is beyond the scenario of bosonic pairing. The rare coexistence of both electrons and holes at the VHS (Fig. 4b) can induce an attractive component of the Coulomb interaction for an unconventional electronic pairing. In this situation, the superconducting state of the $CsV_{2.6}Ta_{0.4}Sb_5$ with a higher $T_c$ would be fundamentally different from that of the $CsV_3Sb_5$ with a lower $T_c$. The van Hove scenario[5] also echoes our STM result that the superconducting state of $CsV_{2.6}Ta_{0.4}Sb_5$ is different from that of the pristine compound.

Our results can also shed new insights on the origin of the CDW order in $CsV_3Sb_5$. The driving mechanism of CDW in $CsV_3Sb_5$ has been primarily attributed to the Fermi surface nesting between VHSs[32,37,40] or electron–phonon coupling[31,41–44]. Our measurements on $CsV_{2.6}Ta_{0.4}Sb_5$ reveal a perfect Fermi surface nesting condition, because the VHS is located exactly at the Fermi level. However, the CDW order is absent in this compound (Fig. 1). These results strongly suggest that the charge order instability is not directly driven by the Fermi surface nesting between the VHSs at M points. Nevertheless, the CDW order and the VHSs are indeed closely related, evidenced by the significantly reconstructed VHSs in the CDW state of $CsV_3Sb_5$[10,11,13,36]. In order to understand this issue, we have carried out first-principles calculations

on the total energy of the $CsV_3Sb_5$ as a function of Ta substitution. In the pristine $CsV_3Sb_5$, the crystal structure with Inverse star of David (ISD) distortion is energetically favorable[37]. However, the total energy of the undistorted kagome structure becomes similar to that of the structure with ISD distortion when the Ta substitution level in $CsV_{3-x}Ta_xSb_5$ reaches $x = 0.25$. Then the undistorted kagome structure becomes energetically most favorable with higher Ta substitution level (see Supplementary Fig. S21). These results have almost quantitatively explained the experimental observation that the CDW order in $CsV_{3-x}Ta_xSb_5$ disappears at high Ta substitution levels with a transition between $x = 0.24$ and $x = 0.3$. As for the origin of the energy change, it would be interesting to consider a cooperative mechanism. In order to lower the total energy of the material system, multiple electronic instabilities are favored by the VHSs, but one of them may dominate if triggered by another interaction. Future experiments are stimulated to verify the proposed mechanism and understand how the tuning knob works for the competing or intertwined orders in kagome superconductors.

## Methods

### Sample growth and characterizations

Single crystals of Ta and Ti substituted $CsV_3Sb_5$ were grown by the self-flux method[6,7]. The crystal structure was examined by X-ray diffraction (XRD) and the element substitution levels were determined by energy-dispersive X-ray spectroscopy (EDS).

### ARPES measurements

Synchrotron-based ARPES measurements were performed at Beamline 5-2 of the Stanford Synchrotron Radiation Lightsource (SSRL) of SLAC National Accelerator Laboratory with a total energy resolution of ~10 meV and a base pressure of better than $3 \times 10^{-11}$ torr. ARPES measurements were also carried out at our lab-based ARPES system with a total energy resolution of ~7 meV and a base pressure of better than $7 \times 10^{-11}$ torr. All samples were cleaved in situ and measured with fresh cleaving surfaces. The Fermi level was determined by measuring the polycrystalline Au in electrical contact with the samples. Unless otherwise noted, the Fermi surface and band structure of the $CsV_3Sb_5$ and Ta substituted $CsV_3Sb_5$ samples were measured with the same experimental conditions for the direct comparison.

### STM measurements

STM measurements were carried out using a customized UNISOKU USM1300 microscope, equipped with a magnetic field up to 11 T that is perpendicular to the sample surface. The electronic temperature is 0.62 K (the energy resolution is about 0.15 meV) at a base temperature of 0.42 K, calibrated using a standard superconductor, Nb crystal. Non-superconducting tungsten tips fabricated via electrochemical etching were annealed in ultrahigh vacuum and calibrated on a clean Au(111) surface. All samples were cleaved in situ at ~10 K and transferred immediately into the STM chambers for measurements. As Sb-terminated surface provides a direct window to study the momentum-resolved structure of the V kagome bands compared to the Cs-terminated surface[14,15], our STM measurements are focused on the large-scale, clean Sb surfaces of both $CsV_{2.6}Ta_{0.4}Sb_5$ and $CsV_3Sb_5$ for comparison. Unless otherwise noted, the $dI/dV$ maps were acquired by a standard lock-in amplifier at a modulation frequency of 973.1 Hz. To remove the effects of small piezoelectric and thermal drifts during the acquisition of $dI/dV$ maps, we apply the Lawler-Fujita drift-correction algorithmon[45], which aligns the atomic Bragg peaks in STM images to be exactly equal in magnitude and 60° apart.

### First-principles calculations

First-principles calculations were performed by using the projected augmented-wave method[46] as implemented in the Vienna ab initio simulation package (VASP)[47,48]. The exchange-correlation interaction

was treated with the generalized gradient approximation (GGA) of the Perdew–Burke–Ernzerhof type[49]. The kinetic energy cutoff and energy threshold for convergence were set to be 520 eV and $10^{-6}$ eV, respectively. The zero-damping DFT-D3 method was used to treat the van der Waals correction[50]. To simulate the Ta substituted system, a $2 \times 2 \times 1$ supercell was constructed, the lattice constants $a = b = 5.4949$ Å, $c = 9.3085$ Å were used (see Supplementary Fig. S22). To directly compare the band structure of the supercell with experiment measurements, we performed band unfolding calculations by using the effective band structure method[51,52] as implemented in VASPKIT codes[53].

## Data availability

The raw data generated in this study are provided in the article and the supplementary materials.

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

## Acknowledgements
We thank K. Jiang, J.-J. Ying, Z.-Y. Wang, T. Wu, and R. S. Markiewicz for useful discussions. We thank X.-H. Han, Y.-H. Ye, J.-Y. Zhao, X.-G. Liu, L.-P. Nie, and T. Wu for their help in the experiments. The work at University of Science and Technology of China (USTC) was supported by the National Natural Science Foundation of China (Nos. 52273309, 12074358, 52261135638, 11974327, and 12004369), the Fundamental Research Funds for the Central Universities (Nos. WK2030000035, WK3510000012, WK3510000010, WK2030020032), the Innovation Program for Quantum Science and Technology (Nos. 2021ZD0302800, 2021ZD0302802), the Anhui Initiative in Quantum Information Technologies (No. AHY170000) and the USTC start-up fund. The work at Beijing Institute of Technology was supported by the National Key R&D Program of China (Grant No. 2020YFA0308800), the Natural Science Foundation of China (Grant No. 92065109), the Beijing Natural Science Foundation (Grant Nos. Z210006, Z190006), and the Beijing Institute of Technology (BIT) Research Fund Program for Young Scholars (Grant No. 3180012222011). Z.W. thanks the Analysis & Testing Center at BIT for assistance in facility support. The work at Institute of Physics was supported by the National Natural Science Foundation of China (61888102, 52022105) and the Key Research Program of Chinese Academy of Sciences (ZDBS-SSW-WHC001). Use of the Stanford Synchrotron Radiation Lightsource, SLAC National Accelerator Laboratory, is supported by the U.S. Department of Energy, Office of Science, Office of Basic Energy Sciences under Contract No. DE-AC02-76SF00515. M.H. and D.L. acknowledge the support of the U.S. Department of Energy, Office of Science, Office of Basic Energy Sciences, Division of Material Sciences and Engineering.

## Author contributions
J.H. and Z.W. initiated the research. J.L., Y.Y., Z.W., H.L., Z.X., and X.C. grew and characterized the crystals. Y.H., S.D., Z.L., and Z.Q. performed the theoretical calculations. H.C., Z.H., H.Y., and H.-J.G. carried out the STM measurements and the QPI simulations. Y.L., L.H., B.W., and J.S. performed the ARPES experiments with guidance from J.H. and help from D.L., M.H., S.P., Z.W., Y.M., X.S., and Z.O. Y.L. analyzed the data. J.H. and Y.L. wrote the paper with inputs from all authors.

## Competing interests
The authors declare no competing interests.
