## [Peer Review File · Nature Communications]

REVIEWERS' COMMENTS

Reviewer #2 (Remarks to the Author):

In the manuscript authors investigate the electronic properties of Kagome metals based on CsV₃Sb₅. Using high-resolution ARPES, STM/STS and first principles calculations, the authors show that Ta and Ti doping into the CsV₃Sb₅ Kagome net suppresses the tendency toward charge density wave (CDW), enhancing the superconducting critical temperature. Moreover, this phenomenon is accompanied with relocation of a van Hove singularity (VHS) precisely at the Fermi level when the maximum of the T_c is reached. The reported STM data suggest that many-body interaction in the superconducting state is rather anisotropic for the doped sample in compare to the pristine one. Authors propose that the saddle nature of the VHS may trigger the T_c enhancement and unusual QPI behavior. Finally based on the total energy calculations authors show how Ta doping change energy balance between different instabilities and suppress the CDW order while the Band structure becomes in favor of CDW phase. Overall data are of exceptional quality and basic conclusions are very well supported by multiple supplementary experiments. The work deserves publication in my opinion.

Minor comment:

Authors stated at the end of introduction part: "We further demonstrate that the suppression of competing orders (e.g. the CDW order) is insufficient to account for the record-high T_c, and the superconducting T_c (gap) is related to the energy position of the VHS in samples without competing orders." While in the main text to explain this point they make a short link to the supplementary information. This makes the main text insufficient to understand the origin of the claim. Thus, some part of the data discussion should be relocated from the SI to the manuscript.

Reviewer #4 (Remarks to the Author):

Appropriate modifications have been made and the strong statements have been toned down. I do not see any major issues with the paper now.

Point-to-Point Response to Referee Reports

For clarity, the referees' original comments are shown by blue italic characters.
The authors' responses are shown by black normal characters.

Referee #2:

In the manuscript authors investigate the electronic properties of Kagome metals based on CsV_3Sb_5 . Using high-resolution ARPES, STM/STS and first principles calculations, the authors show that Ta and Ti doping into the CsV_3Sb_5 Kagome net suppresses the tendency toward charge density wave (CDW), enhancing the superconducting critical temperature. Moreover, this phenomenon is accompanied with relocation of a van Hove singularity (VHS) precisely at the Fermi level when the maximum of the T_c is reached. The reported STM data suggest that many-body interaction in the superconducting state is rather anisotropic for the doped sample in compare to the pristine one. Authors propose that the saddle nature of the VHS may trigger the T_c enhancement and unusual QPI behavior. Finally based on the total energy calculations authors show how Ta doping change energy balance between different instabilities and suppress the CDW order while the Band structure becomes in favor of CDW phase. Overall data are of exceptional quality and basic conclusions are very well supported by multiple supplementary experiments. The work deserves publication in my opinion.

We thank the referee for reviewing our work again and recommending the publication of our paper in Nature Communications.

Minor comment:

Authors stated at the end of introduction part: "We further demonstrate that the suppression of competing orders (e.g. the CDW order) is insufficient to account for the record-high T_c , and the superconducting T_c (gap) is related to the energy position of the VHS in samples without competing orders." While in the main text to explain this point they make a short link to the supplementary information. This makes the main text insufficient to understand the origin of the claim. Thus, some part of the data discussion should be relocated from the SI to the manuscript.

We thank the referee for the constructive suggestion. We have followed the referee's suggestion and moved the relevant part of the data discussion from the supplementary materials to the main manuscript.

Referee #4:

Appropriate modifications have been made and the strong statements have been toned down. I do not see any major issues with the paper now.

We thank the referee for reviewing our work again.

Summary of changes:

1. As suggested by referee #2, we have now moved the relevant part of the data discussion from the supplementary materials to the main manuscript.
2. We have made the modifications suggested by the editor.